# Current Biomarkers for Alzheimer’s Disease: From CSF to Blood

**DOI:** 10.3390/jpm10030085

**Published:** 2020-08-12

**Authors:** Kun Zou, Mohammad Abdullah, Makoto Michikawa

**Affiliations:** Department of Biochemistry, Nagoya City University Graduate School of Medical Sciences, Kawasumi 1, Mizuho-cyo, Mizuho-ku, Nagoya 467-8601, Aichi, Japan; mabdullahz49@gmail.com (M.A.); michi@med.nagoya-cu.ac.jp (M.M.)

**Keywords:** Alzheimer’s disease, biomarker, cerebrospinal fluid, blood

## Abstract

Alzheimer’s disease (AD) is the most common cause of dementia and affects a large portion of the elderly population worldwide. Currently, a diagnosis of AD depends on the clinical symptoms of dementia, magnetic resonance imaging to determine brain volume, and positron emission tomography imaging to detect brain amyloid or tau deposition. The best characterized biological fluid markers for AD are decreased levels of amyloid β-protein (Aβ) 42 and increased levels of phosphorylated tau and total tau in cerebrospinal fluid (CSF). However, less invasive and easily detectable biomarkers for the diagnosis of AD, especially at the early stage, are still under development. Here, we provide an overview of various biomarkers identified in CSF and blood for the diagnostics of AD over the last 25 years. CSF biomarkers that reflect the three hallmarks of AD, amyloid deposition, neurofibrillary tangles, and neurodegeneration, are well established. Based on the need to start treatment in asymptomatic people with AD and to screen for AD risk in large numbers of young, healthy individuals, the development of biomarkers for AD is shifting from CSF to blood. Elements of the core pathogenesis of AD in blood, including Aβ42, tau proteins, plasma proteins, or lipids have shown their usefulness and capabilities in AD diagnosis. We also highlight some novel identified blood biomarkers (including Aβ42/Aβ43, p-tau 181, Aβ42/APP669-711, structure of Aβ in blood, and flotillin) for AD.

## 1. Introduction

Alzheimer disease (AD) is an age-dependent neurodegenerative disorder and the most prevalent form of dementia in the elderly population. AD is characterized by amyloid β-protein (Aβ) deposition in senile plaques in the brain parenchyma and by phosphorylated tau deposition in neurofibrillary tangles in cerebral neurons [1]. Until the 2000s, clinical diagnosis of AD depended on clinical symptoms, cognitive examination, and the exclusion of other etiologies of dementia. A definitive positive diagnosis of AD could only be made by post-mortem pathological confirmation of brain parenchymal Aβ deposition and neurofibrillary tangles [2]. Later, structural imaging of the hippocampus with magnetic resonance imaging became an integral part of the clinical assessment of patients with AD [3,4]. Recently, innovative imaging for brain Aβ deposition in patients using positron emission tomography (PET) technology was approved for clinical use [5].

The exploration of AD biomarkers in biological fluid has focused on the core molecules of AD pathogenesis, Aβ and tau proteins. Aβ in brain senile plaques contains 40–43 amino acids, and Aβ42 and Aβ40 are the major species generated by sequential proteolytic cleavage of amyloid precursor protein (APP) by β- and γ-secretases [6]. Most APP undergoes non-amyloidogenic processing by α-secretase, generating a non-amyloidogenic fragment of Aβ, called p3 [7]. The longer species, Aβ42 and Aβ43, are highly prone to aggregation, deposit early in the brain, and the oligomers are highly toxic to neurons [1,8,9], whereas Aβ40 may have antioxidant and anti-amyloidogenic effects [10,11,12]. Although numerous arguments remain regarding the causative molecule for AD, Aβ or phosphorylated tau (p-tau), extensive evidence suggests that Aβ42 deposition in the brain parenchyma appears earlier than p-tau deposition in neurofibrillary tangles and can be detected in the brain many years prior to the appearance of AD clinical symptoms [13]. In addition, the strongest evidence for Aβ42 as the causative molecule for AD is from genetic studies of familial AD (FAD) with mutations in *A*PP, presenilin 1 (P*SEN1*, PS), or presenilin 2 (P*SEN2*, PS), which lead to the highest risk for AD among all AD risk genes identified so far [14]. PS1 and PS2 are the catalytic components of γ-secretase for Aβ generation. The involvement of either substrate (APP) or enzyme (PS) in FAD indicates a central role for Aβ42 in AD pathogenesis [15]. Thus, Aβ42 became the most important target in terms of both biomarkers and therapeutic strategy development for AD.

Current approved treatments for AD target its symptoms, but more and more clinical trials are testing potential disease-modifying drugs, which target the most upstream molecule of AD pathogenesis: Aβ42. However, over the past few decades, many anti-Aβ clinical trials have failed to treat symptomatic AD. Recently, Phase 3 clinical trials using a β-secretase inhibitor or a γ-secretase inhibitor to inhibit Aβ generation, or using Aβ antibodies to promote Aβ clearance in the early or mid-stage of AD, also failed to achieve their expected therapeutic effect [16,17,18]. One suggestion for these failures is that the anti-Aβ treatment could not rescue degenerative neurons or synapses that were already damaged by toxic Aβ42. Current clinical trials targeting Aβ have focused on preclinical patients without AD symptoms rather than symptomatic AD patients or patients with mild cognitive impairment (MCI). Thus, reliable biomarkers are required for rapid, early, and less-invasive detection in the predementia phase. Biomarkers identified from cerebrospinal fluid (CSF) and blood have shown high potential to diagnose AD at the early stage or to predict AD onset in the future.

In this mini review, we highlight biomarkers for AD diagnosis found in CSF and blood, including the core pathological proteins, Aβ42 and tau, and neurodegeneration- and metabolism-related biomarkers (Figure 1). In addition to the breakthrough finding of decreased Aβ42 and increased p-tau in CSF for AD diagnosis, additional, less-invasive, and easily accessible blood biomarkers are emerging.

## 2. CSF Markers of AD

### 2.1. AD Pathogenesis Molecule-Based Biomarkers in CSF

#### 2.1.1. CSF Aβ Markers

CSF is in indirect contact with the extracellular space of the brain, and biochemical changes in the brain are therefore reflected in the CSF. Neurovascular and blood–brain barrier dysfunction may develop in neurodegenerative diseases. CSF is thus the optimal source for AD biomarkers [19]. The core pathogenic molecules for AD, Aβ42, total tau (t-tau), and p-tau in CSF, were studied beginning in the 1990s and have become reliable and sensitive biomarkers for AD diagnosis [20,21]. A marked decrease in CSF Aβ42 and a marked increases in CSF t-tau and p-tau can be used to identify symptomatic AD patients with a sensitivity and specificity above 80% [22]. The decrease in CSF Aβ42 is also useful for predicting later AD development. In a population-based study, Skoog et al. found that CSF Aβ42 is reduced before the onset of sporadic dementia [23]. Gustafson et al. reported that low levels of CSF Aβ42, but not high t-tau, may predict cognitive decline in older women [24]. Similarly, Stomrud et al. found that CSF Aβ42, but not t-tau or p-tau, can predict cognitive decline in healthy elderly people [25]. In contrast to CSF Aβ42, CSF Aβ40, the predominant species of Aβ, did not show a significant change in an AD group compared with a normal control group [20]. In another study by Nutu et al., the CSF Aβ40 level in AD was higher than that in Parkinson’s disease dementia (PDD) and dementia with Lewy bodies (DLB), suggesting that the Aβ42/40 ratio may improve differentiation of AD from PDD and DLB [26]. Janelidze et al. also found that decreases in the Aβ42/40 and Aβ42/38 ratios may be better diagnostic markers of AD than CSF Aβ42 alone for discrimination of AD from non-AD conditions, especially from DLB, PDD, and vascular dementia (VaD) [27]. Similarly, Baldeiras et al. reported that addition of the Aβ42/40 ratio to the CSF biomarker profile increases the predictive value for underlying AD in MCI [28]. Recently, the usefulness of using CSF Aβ42 to predict preclinical AD was confirmed in cognitively normal individuals with inherited AD genes. An FAD cohort study from the Dominantly Inherited Alzheimer Network indicated that CSF Aβ42 levels may first increase and then start to decline 25 years before the onset of AD symptoms, whereas amyloid deposition measured with PET and Pittsburgh compound B and an increased concentration of t-tau in the CSF can be detected 15 years before expected symptom onset [29]. These findings raise the possibility that the decline in CSF Aβ42 may be the earliest marker for identifying preclinical AD, not only in FAD, but also in sporadic AD. However, obtaining CSF is invasive, risky, and unsuitable for screening healthy people.

Combined with hippocampal volumetry, fluorodeoxyglucose (FDG) PET, amyloid PET, decreased Aβ42, and increased tau or p-tau in CSF have been used in the National Institute for Aging-Alzheimer Association (NIA-AA) criteria to diagnose AD, to predict later onset of AD, and to differentiate AD from normal aging with MCI [1].

Oligomerization of Aβ42 has long been suggested as the central pathogenic event in AD [6,30]. The decrease in CSF Aβ42 was thought to be caused, at least in part, by deposition in amyloid plaques. Another interpretation is that oligomerization or aggregation of CSF Aβ42 reduces the detection of Aβ42 monomers with an enzyme-linked immunosorbent assay (ELISA). One ELISA method for examining Aβ oligomers was designed by Fumumoto et al., using the same antibody for capture and detection. Monomeric Aβ will not be detected because the epitope is already occupied by the captured antibody. They found that the level of Aβ oligomers in CSF was higher in AD or MCI patients compared with age-matched controls [31]. However, this study cannot identify the nature of the Aβ oligomers, e.g., dimers, trimers, or tetramers, and this finding needs to be confirmed in other larger, independent studies.

#### 2.1.2. CSF t-tau and p-tau Markers

Tau proteins are located in neuronal axons and play roles in maintaining the stability of microtubules in neurons of the central nervous system (CNS). Tau proteins in the human brain are composed of six soluble isoforms and numerous phosphorylation sites [32]. Hyperphosphorylated tau proteins disassociate from microtubules and form insoluble aggregates in neurons, which are called neurofibrillary tangles [33,34]. CSF t-tau and p-tau are frequently studied in neurodegenerative diseases. CSF t-tau levels can serve as a neuronal injury marker and are increased in many neurodegenerative diseases, such as Creutzfeldt-Jakob disease [35], AD, DLB, and frontotemporal dementia (FTD), whereas CSF p-tau 181 or p-tau 231 (tau phosphorylated at threonine 181 or threonine 231) levels increase more specifically in AD than in other neurodegenerative diseases. Thus, p-tau may reflect the hyperphosphorylation of tau and the formation of neurofibrillary tangles in AD [2,36,37].

Blennow et al. demonstrated that marked elevations of t-tau and paired helical filaments (PHF)-tau (tau phosphorylated at serine 202 and threonine 205) are consistently found in the CSF of AD patients. However, moderate elevations of t-tau and PHF-tau are also found in other neurodegenerative diseases, such as VaD and frontal lobe dementia [38]. Later, Vanmechelen et al. reported a method using sandwich ELISA for detecting p-tau 181 and found that CSF p-tau 181 levels were increased in AD patients compared with age-matched controls, whereas levels were decreased in patients with FTD, suggesting that CSF p-tau 181 could be a more specific marker for AD [39]. Kohnken et al. developed a sandwich ELISA for detecting p-tau threonine 231 that shows 85% sensitivity and 97% specificity for discrimination of AD from non-AD controls [40]. These findings were confirmed in numerous subsequent studies. In a meta-analysis comprised of 164 cohorts with AD and 153 control cohorts representing 11,341 AD patients and 7086 controls, increased levels of CSF t-tau and p-tau were strongly associated with AD and MCI patients that developed AD [41]. Similar to CSF Aβ42, although CSF t-tau and p-tau are already included in the diagnostic criteria for symptomatic or prodromal AD, they are difficult to use in healthy people at the preclinical stage because of the limitation of obtaining CSF samples.

#### 2.1.3. CSF β-Site APP-Cleaving Enzyme 1 (BACE1) Marker

BACE1 is the major β-secretase responsible for Aβ generation. Mutations in *BACE1* have not yet been reported in FAD. However, mutations in APP near the β-cleavage site may be responsible for early-onset FAD (Swedish mutation) or may be protective for preventing late-onset sporadic AD [42]. To study whether up-regulation of BACE1 is an early pathogenic event in AD, some human in vivo studies have reported good diagnostic performance of CSF BACE1 levels and activity for separating symptomatic AD patients and patients with MCI from cognitively healthy controls. Holsinger et al. found increased BACE1 activity in the CSF of AD patients, whereas Zhong et al. showed that increased CSF BACE1 levels can be a predictor of risk in patients with MCI [43,44]. Later, Ewers et al. reported that increased CSF BACE1 activity is not only associated with the *a*p*oE4* genotype in MCI and AD patients, but is also associated with decreased hippocampal volume in AD [45,46]. However, the diagnostic value of CSF BACE1 activity requires further evaluation and confirmation in larger studies from different groups.

### 2.2. Neurodegeneration-Based Biomarkers in CSF

In addition to the three core CSF biomarkers, Aβ42, t-tau, and p-tau, biomarkers that reflect axonal neurodegeneration, synapse loss, and activation of glial cells have also been extensively explored [37,47].

Neurofilaments are intermediate filaments that serve as structural components of neuronal axons, particularly large myelinated axons. In the CNS, neurofilaments are heteropolymers and are composed of four subunits, neurofilament light (NF-L), neurofilament middle, neurofilament heavy, and α-internexin [48]. NF has been extensively examined in patients with neuronal injury and neurodegenerative diseases because it is released into CSF and systemic circulation when neurons are injured [49]. Recently, Sjogren et al. found that CSF NF-L levels are increased in patients with FTD and late onset AD compared with control subjects, and the increase in FTD patients is higher than in late onset AD [50]. In a meta-analysis, Olsson et al. found that NF-L has a large effect size for differentiating AD patients from control individuals [41]. However, high CSF levels of NF-L are also found in other neurodegenerative diseases, such VaD, normal-pressure hydrocephalus, multiple sclerosis, and amyotrophic lateral sclerosis [51,52,53,54]. Thus, CSF NF-L could be a representative marker of neurodegeneration, but not a specific marker for distinguishing AD from other neurological disorders. Nevertheless, Zetterberg et al. showed that higher CSF NF-L concentrations are associated with cognitive deterioration and brain atrophy over time in AD and MCI groups, indicating that CSF NF-L can be used as a marker for AD progression [49].

Visinin-like protein 1 (VILIP-1) is a neuronal calcium sensor protein that is exclusively expressed in neurons and can be used as a brain injury marker [55]. Lee et al. found that CSF VILIP-1 levels are significantly higher in AD patients compared with controls and that the diagnostic performance of VILIP-1 is similar to CSF Aβ42, t-tau, or p-tau [56]. Higher CSF VILIP-1 levels in AD patients compared with controls have also been reported in several other studies. Tarawneh et al. reported that CSF VILIP-1 and CSF VILIP-1/Aβ42 ratios are increased in early AD, suggesting the utility of them as diagnostic or prognostic markers of AD [57]. Later, they reported that CSF VILIP-1 levels can predict rates of whole-brain and regional atrophy, similar to tau and p-tau 181 [58]. CSF VILIP-1 levels have been consistently shown to be higher in AD patients compared with normal controls [59,60,61]. Luo et al. showed that CSF VILIP-1 levels are significantly increased in AD patients compared with both normal controls and DLB patients. However, a recent meta-analysis performed by Mavroudis et al. did not show a significant difference between AD and DLB [62]. Because the reports are limited, whether CSF VILIP-1 can be used as a specific marker for AD that discriminates AD from other neurodegenerative diseases needs to be further studied.

In addition to NF-L, which represents axonal damage, several pre- and postsynaptic proteins are increased in the CSF of AD patients, such as neurogranin, synaptosome-associated protein 25 (SNAP-25), and synaptotagmin. Neurogranin is a postsynaptic protein that is predominantly expressed in dendritic spines and plays a role in postsynaptic signaling pathways. Using immunoprecipitation enrichment of neurogranin, Thorsell et al. found a significant increase in neurogranin in the CSF of AD patients compared with a control group [63]. Several studies from different groups consistently confirmed higher levels of CSF neurogranin in AD patients compared to controls [64,65,66]. Keter et al. further showed that CSF levels of neurogranin are higher in patients with MCI who progressed to AD compared with those with stable MCI, indicating that neurogranin can be used as a predictive factor of progression from MCI to AD [67]. Tarawneh et al. proposed the diagnostic and prognostic utility of CSF neurogranin as a synaptic marker in early symptomatic AD [68]. However, the diagnostic value of CSF neurogranin in AD or MCI is still based on other diagnostic indexes of AD. Lista et al. have shown that CSF neurogranin concentrations are significantly higher in AD patients compared with FTD patients [66]. CSF neurogranin levels in other types of dementia or neurodegenerative diseases need to be studied further.

By using novel affinity mass spectrometry, Brinkmalm et al. found significantly higher levels of CSF SNAP-25 fragments in AD patients than controls [69]. In another longitudinal study, Sutphen et al. also revealed that CSF SNAP-25 levels are significantly higher in AD and MCI patients than controls, but decline over time in the AD group [61]. The increase in SNAP-25 fragments in CSF has the highest power among synaptic biomarkers to distinguish AD patients from non-AD patients [70].

Ohrfelt et al. reported that the presynaptic protein, synaptotagmin, is significantly increased in the CSF of patients with AD, or MCI patients that developed AD [71]. Recently, Tible et al. confirmed that all these synaptic biomarkers are significantly increased in patients with AD and MCI patients that developed AD. Given that the synaptic proteins are general markers of synaptic dysfunction, they likely can be used as supplementary diagnostic biomarkers for AD or MCI patients that have developed AD, but not as exclusive diagnostic markers for AD.

## 3. Blood Markers of AD

### 3.1. AD Pathogenesis Molecule-Based Biomarkers in Blood

#### 3.1.1. Blood Aβ Markers

As with CSF Aβ levels, Aβ42 and Aβ40 are the most extensively studied blood markers for the diagnosis of symptomatic and prodromal AD. During the first decade of the 2000s, the findings regarding Aβ42 and Aβ40 levels in the plasma of AD patients were not consistent, and sometimes the results were contradictory. Mayeux et al. found increased plasma Aβ42 levels, but not plasma Aβ40, in AD patients at baseline and in those who developed AD within 3 years in a follow-up study. The risk of AD onset in individuals with high plasma Aβ42 was increased more than 2-fold compared to those with low plasma Aβ42 [72]. In later studies, van Oijen et al. reported that a high concentration of plasma Aβ40 is associated with an increased risk of dementia [73], whereas Yaffe et al. found that a lower plasma Aβ42/Aβ40 ratio is associated with greater cognitive decline among elderly persons without dementia over 9 years [74]. This discrepancy may come from the clinical stage of examination and/or the mix of other types of dementia.

Using magnetic resonance imaging for hippocampal volumetry and amyloid PET technology, patients with AD and MCI patients that developed AD can be specifically discriminated from patients with other types of dementia and MCI patients that did not develop AD in the last decade. Zou et al. showed that Aβ42 levels are lower, and Aβ43 levels are higher, in the serum of AD patients compared with age-matched normal controls, suggesting that a lower Aβ42/Aβ43 ratio can be used as a blood marker for AD diagnosis [9]. In two independent data sets, Nakamura et al. also revealed a significant decrease in plasma Aβ42 levels in brain amyloid-positive AD or MCI patients compared with cognitively normal individuals. They also found that the combination of decreased Aβ42/APP669–711 and the Aβ42/Aβ40 ratio showed the highest and most stable performance in predicting brain amyloid burden at an individual level [75]. Recently, a correlation between a lower plasma Aβ42/Aβ40 ratio and amyloid burden was consistently confirmed in other independent studies. Perez-Grijalba et al. showed that a decreased plasma Aβ42/Aβ40 ratio alone can accurately predict positivity and detect early stages of AD [76,77]. Using a multiplex sensor array, Kim et al. showed that a lower Aβ42/Aβ40 ratio and a higher plasma t-tau/Aβ42 and p-tau 181/Aβ42 ratios successfully discriminated AD patients from healthy controls [78]. In addition to plasma Aβ levels, Nabers et al. showed that a change in the secondary structure of Aβ in human blood plasma can be used as a blood amyloid indicator for prodromal AD. The change to an increased β-sheet structure of Aβ is correlated with CSF AD biomarkers and amyloid PET imaging [79]. Because the structure of Aβ is not stable and may change in several hours, this technology needs relatively high techniques and must be confirmed by other independent studies.

Consistent with the findings of lower CSF Aβ42 levels in AD patients, recent studies strongly suggest that plasma Aβ42 levels also decrease in AD patients or amyloid-positive MCI patients. Thus, the combined use of Aβ42/Aβ40, Aβ42/Aβ43, Aβ42/APP669-711, Aβ42/t-tau, or Aβ42/p-tau 181 may accurately diagnose or predict AD.

#### 3.1.2. Blood p-tau Markers

Because of the invasiveness and high costs of examining CSF tau, plasma tau has also become a candidate blood marker for AD diagnosis, and many studies have focused on quantitation of tau in AD, MCI, and normal groups. Because the tau levels in plasma are much lower than in CSF, an ultra-sensitive assay was developed by Zetterberg et al. They found elevated t-tau levels in plasma from patients with AD compared with those from control or MCI patients, whereas no difference was found between MCI patients that developed AD and stable MCI patients [80]. In a later study, Mattsson et al. studied two large cohorts and reported that plasma t-tau may partly reflect AD pathology, but a large overlap was found between patients with AD and age-matched controls, suggesting that using plasma t-tau as an AD biomarker in individual people is difficult [81].

Recently, Tatebe et al. developed a novel ultrasensitive immunoassay for the quantitation of plasma p-tau 181. Although the number of participants was small, they clearly showed that plasma p-tau 181 is significantly increased in patients with AD, as well as in patients with Down’s syndrome, compared with normal controls [82]. Karikari et al. further confirmed the increase in plasma p-tau 181 levels in patients with AD and MCI patients that developed AD and showed that plasma p-tau 181 can discriminate AD dementia from not only normal young and older adults, but also FTD, VaD, progressive supranuclear palsy, corticobasal syndrome, Parkinson’s disease, and multiple system atrophy [83].

### 3.2. Other Biomarkers in Blood

Because neuronal or synaptic biomarkers indicate general neuronal injury in many neurodegenerative diseases, the blood levels of these proteins may not be specific markers for AD. Benussi et al. assessed the diagnostic and prognostic value of serum NF-L and serum p-tau 181. They found that serum NF-L levels are increased in both FTD and AD and cannot distinguish AD from FTD, whereas serum p-tau 181 levels are specifically increased in patients with AD [84]. Similarly, plasma NF-L is increased in both progressive supranuclear palsy and AD [85]. Regarding the use of other neurodegeneration-based biomarkers for AD diagnosis such as VILIP-1, neurogranin, SNAP-25, and synaptotagmin, very few studies were performed on their levels in plasma, and the results were inconsistent and very limited [37,47]. Therefore, these neuronal and synaptic biomarkers are currently considered to be representative of neurodegeneration, and may be parameters for assessing the progression or degree of AD and other types of dementia, but not useful for accurate diagnosis of AD.

In addition to the molecules related to AD core pathogenesis and neurodegeneration, other plasma proteins, lipids, and metabolites were also extensively studied in patients with AD. Ray et al. used ELISA and identified 18 signaling proteins in blood plasma. The change in the pattern of those proteins can distinguish AD and MCI that progressed to AD from control subjects with near 90% accuracy [86]. Using a multiplex assay, Doecke et al. identified another set of plasma proteins that distinguishes individuals with AD from healthy controls with high sensitivity and specificity [87], and Hye et al. identified 10 plasma proteins that are strongly associated with progression from MCI to AD [88]. The change in these plasma proteins seems to result from neurodegeneration or other systemic disorders in AD. Because the number of these plasma proteins is large, and examining all of the proteins is expensive, the patterns of change in these proteins in AD have still not been confirmed by other independent studies.

The systemic abnormalities in lipid metabolism in the blood of AD patients have also been studied by using quantitative and targeted metabolomics and mass spectrometry. Mapstone et al. identified 10 phospholipids from healthy elderly people that predicted conversion to either MCI or AD within 2–3 years with over 90% accuracy, suggesting their use in detection of early neurodegeneration in preclinical AD [89]. Recently, Varma et al. also identified four sphingolipids and found that their higher blood concentrations in cognitively normal individuals are associated with an increased risk of future conversion to incident AD [90]. The change in the levels of these phospholipids and sphingolipids in blood may reflect a disorder of lipid metabolism and/or neuronal degeneration in the CNS at the very early stage without cognitive symptoms. However, whether they distinguish AD from other types of dementia and neurodegenerative diseases needs to be further investigated. Given the high cost of quantifying a set of plasma proteins or lipids, single blood markers could be easier to use for screening for AD in large populations. In a recent study, Abdullah et al. identified flotillin, an abundant exosome protein, as a novel diagnostic marker for AD. Serum flotillin levels are significantly decreased in patients with AD and amyloid-positive MCI patients compared with age-matched patients with VaD and MCI patients without amyloid [91]. The decrease in flotillin levels may result from reduced exosome secretion caused by Aβ42 oligomers [92]. Thus, flotillin is likely to be a secondary responding molecule to pathogenetic Aβ42. The above CSF and blood biomarkers for AD were summarized in Table 1.

## 4. Advantages of Blood Biomarkers over CSF Biomarkers

CSF biomarkers for AD have been studied for more than 20 years, and many powerful markers have been identified for diagnosis, prognosis, or even prediction of the future onset of AD. The combined use of these CSF markers may largely improve the accuracy and sensitivity of AD diagnosis at the early stage. However, because obtaining CSF is invasive and may induce prognostic symptoms, physical examination using CSF samples to screen for the risk of AD in large populations of asymptomatic people is not practical.

Blood test indexes, such as blood cholesterol, triglyceride, high-density lipoprotein cholesterol, and low-density lipoprotein cholesterol, have been widely used for predicting the risks for arteriosclerosis and cerebrovascular and cardiovascular diseases in healthy and asymptomatic populations. However, a safe, less invasive, and readily accessible blood marker for AD diagnosis or for predicting the risk of AD is still not at clinical use stage. In annual physical examinations, blood samples were routinely collected in a large and healthy population from middle age to advanced age. The number and scale of blood samples will take great advantage over CSF samples of developing blood biomarkers for AD. In the past 15 years, many studies on blood markers for AD diagnosis, prognosis, and prediction have been performed, and some biomarkers have emerged as candidates for less invasive blood markers for AD (Table 1). For example, recent studies from different groups suggest that decreased blood Aβ42 and increased blood p-tau 181 may reflect brain amyloid deposition and neurofibrillary tangles, respectively, at the early stage of AD. The changes of some plasma proteins, lipids, Aβ43, and flotillin in the blood samples from AD patients are also needed to be confirmed by different groups. Nevertheless, using blood-borne biomarkers to make a clear AD diagnosis or prognosis will be available in the near future.

## 5. Conclusions

In this first quarter of the century, hundreds of biomarkers aiming for AD diagnosis and for the early detection of pathological changes in AD have been investigated and reported. Biomarkers reflecting the three hallmarks of AD, amyloid deposition, neurofibrillary tangles, and neurodegeneration, have shown a high accuracy in assisting with AD diagnosis. Of all the biomarkers, CSF biomarkers, including decreased Aβ42 and increased t-tau and p-tau, have been well-established for AD diagnosis and the prediction of future conversion to AD from MCI. These core pathogenesis markers of AD have been included in the diagnostic criteria of AD in NIA-AA; however, the invasiveness of obtaining CSF largely limits their utility in cognitively normal populations. Neurodegeneration-based markers in CSF, including NF-L, VILIP-1, neurogranin, and SNAP-25, also showed high positive correlations with neuronal damage in AD and MCI and can be used for evaluation and prediction of future cognitive decline in AD. Of note, these neurodegeneration markers change when neural damage occurs in various neurodegenerative diseases. Thus, they can be auxiliary markers, especially for evaluating the degree of neuronal damage in AD, but may not be suitable for differential diagnosis of AD dementia from other types of dementia.

In addition to CSF biomarkers, blood markers for AD diagnosis and prediction have been extensively studied. Although some contradictory results were reported regarding the blood levels of Aβ42 in AD patients, recent studies showed decreased blood Aβ42 levels in amyloid-positive AD and MCI patients. Furthermore, small but detectable amounts of p-tau and t-tau are increased in the plasma of AD and MCI patients. However, considerable overlap exists in the plasma Aβ42, t-tau, and p-tau levels between AD patients and age-matched controls. Further identification of other potential molecules and use of the ratios of these molecules to Aβ42 or tau proteins may significantly improve the accuracy and sensitivity for screening and discriminating prodromal or preclinical AD from the normal population. Some sets of plasma proteins and lipids may also have potential in AD diagnosis; however, more specific biomarkers are needed and the cost of examination needs to decrease.

Because effective drugs to stop the progression of AD are still not available, preventive therapies and disease-modifying treatments need to be started at the preclinical stage. Discovering new targets for early AD diagnosis and therapy is still necessary in the future direction of AD research. To screen for a risk of AD in healthy populations, the development of AD biomarkers has shifted to using less invasive (blood) or non-invasive (saliva or urine) samples. Given the extensive studies and convincing evidence provided for blood biomarkers, they are likely to be the next generation of biomarkers for AD diagnosis and risk screening.

## Figures and Tables

**Figure 1 jpm-10-00085-f001:**
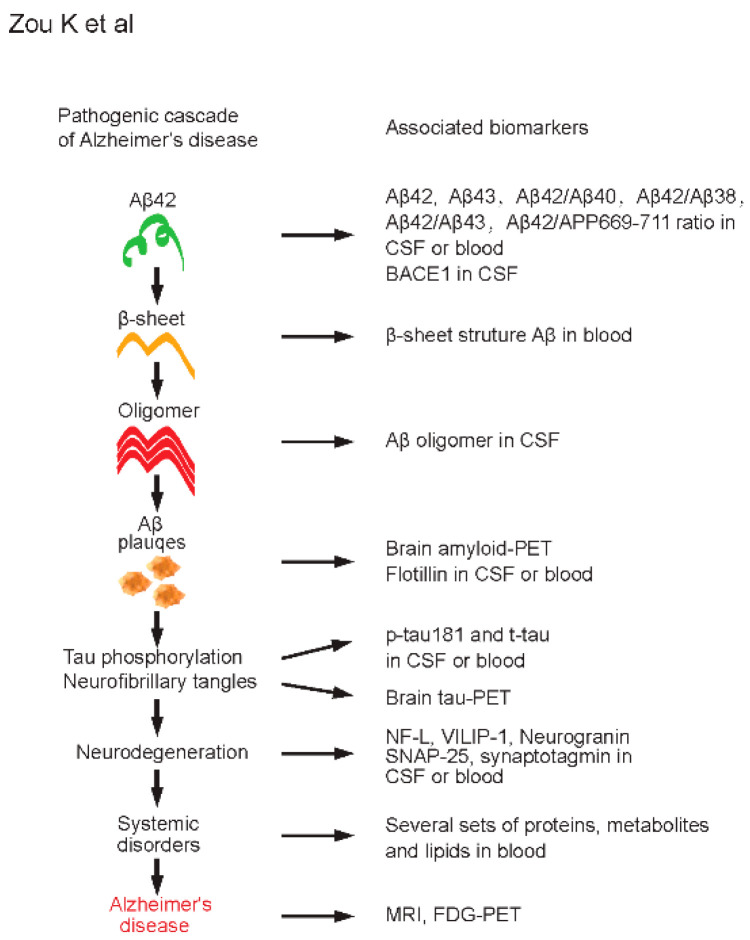
Pathogenic cascade and associated biomarkers of Alzheimer’s disease (AD). Amyloid cascade hypothesis of AD is shown on the left side. Selected and associated biomarkers at each pathogenic stage of AD are shown on the right side. Aβ42, amyloid β-protein 42; Aβ43, amyloid β-protein 43; Aβ40, amyloid β-protein 40; Aβ38, amyloid β-protein 38; APP, amyloid precursor protein; BACE1, β-site APP-cleaving enzyme 1; CSF, cerebrospinal fluid; PET, positron emission tomography; p-tau, phosphorylated tau; t-tau, total tau; NF-L, neurofilament light; VILIP-1, visinin-like protein 1; SNAP-25, synaptosome-associated protein 25; MRI, magnetic resonance imaging; FDG, fluorodeoxyglucose.

**Table 1 jpm-10-00085-t001:** Selected biomarkers of AD in cerebrospinal fluid (CSF) and blood.

Biomarker	Relevance in AD	Change in CSF/Blood of AD
Aβ42	Distinguishing AD, mild cognitive impairment (MCI) that developed AD and preclinical AD from normal controls and other neurodegenerative disease	Consistently decreased in CSF, also decreased in blood [20,21,22,23,24,25,26,27,28,29,74,75,76,77,78]
Aβ40	Inconsistent results for Aβ40 alone, Aβ42/Aβ40 ratio could be a better biomarker than Aβ42 alone	Aβ42/Aβ40 ratio consistently decreased in CSF, also decreased in blood [20,21,22,23,24,25,26,27,28,29,73]
Aβ38	Inconsistent results for Aβ38 alone, Aβ42/Aβ38 ratio could be a better biomarker than Aβ42 alone for discrimination of AD from other dementia	Aβ42/Aβ38 ratio decreased in CSF, very few studies [27]
Aβ43	Distinguishing AD from normal controls	Aβ43 increased and Aβ42/Aβ43 ratio decreased in blood, very few studies [9]
Aβ42/APP669-711	Distinguishing AD from normal controls and MCI that developed AD	Decreased in blood, one study [75]
BACE1	Distinguishing AD and MCI that developed AD from normal controls	Activity and levels increased in CSF, few studies [43,44,45,46]
β-sheet structure Aβ	Correlated with amyloid-PET and other established CSF AD biomarkers	Increased in blood, one study [79]
Aβ oligomer	Distinguishing AD and MCI that developed AD from normal controls	Increased in CSF, very few studies [31]
Flotillin	Distinguishing AD and MCI that developed AD from normal controls and vascular dementia (VaD); single blood marker	Decreased in CSF and blood, very few studies [91]
p-tau and t-tau	Distinguishing AD and MCI that developed AD from normal controls, p-tau 181 and p-tau 231 discriminates AD from other dementia	Consistently increased in CSF [38,39,40,41]; p-tau 181 increased in blood, several studies [80,81,82,83]
NF-L	Distinguishing AD from normal controls, but not other dementia; valuable for assessing neuronal injury	Increased in CSF and blood, several studies [41,49,50,51,52,53,54,84,85]
VILIP-1	Distinguishing early AD and AD from normal controls, but not other dementia	Increased in CSF, inconsistent and limited results in blood [37,47,55,56,57,58,59,60,61,62]
Synaptic proteins (neurogranin, SNAP-25, synaptotagmin)	Distinguishing AD and MCI developed to AD from normal controls, but not other dementia	Increased in CSF, inconsistent and limited results in blood [37,47,61,63,64,65,66,67,68,69,70,71]
18 Signaling proteins	Distinguishing AD and MCI developed to AD from normal controls	Pattern changed in blood, very few studies [86]
10 plasma proteins	Predicting progression from MCI to AD	Pattern changed in blood, very few studies [87,88]
10 phospholipids	Detecting preclinical AD from normal controls	Pattern changed in blood, very few studies [89]
4 sphingolipids	Detecting prodromal and preclinical AD from normal controls	Increased in blood, very few studies [90]

Aβ42, amyloid β-protein 42; MCI, mild cognitive impairment; Aβ40, amyloid β-protein 40; Aβ38, amyloid β-protein 38; Aβ43, amyloid β-protein 43; APP, amyloid precursor protein; CSF, cerebrospinal fluid; p-tau, phosphorylated tau; t-tau, total tau; VaD, vascular dementia; NF-L, neurofilament light; VILIP-1, visinin-like protein 1; SNAP-25, synaptosome-associated protein 25.

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
