# Peer review of "Current Biomarkers for Alzheimer’s Disease: From CSF to Blood"

_jpm, 2020, doi:10.3390/jpm10030085_

Round 1

Reviewer 1 Report

The review entitled “Current biomarkers for Alzheimer’s disease: from CSF to blood” is a well written article focusing on all the biomarkers involved in AD pathogenesis so far and the benefits of their use. However, there are a few suggestions that could improve the overall quality of the manuscript.

  1. Authors can include a table enumerating all the present biomarkers and their relevance/function in Alzheimer’s disease along with the references to studies based on them for a better readability.
  2. A separate section highlighting the advantages of using blood-borne biomarkers over the CSF ones, as mentioned by authors in abstract, would help in an improved understanding of the main idea behind this review.
  3. Have there been any clinically relevant study with blood biomarkers in AD that can be added in the text to highlight possible use of blood biomarkers in a clear AD diagnosis and treatment since all others have failed till now?
  4. A simple figure describing the disease pathology of AD in terms of amyloid precursor protein processing, Aβ aggregation, tau neurofibrillary tangles and secretory components involved in the diseases etiology would increase the overall value of the manuscript.

Author Response

Comment 1

Authors can include a table enumerating all the present biomarkers and their relevance/function in Alzheimer’s disease along with the references to studies based on them for a better readability.

Our response

We have included a table (Table 1) to the present the biomarkers discussed in this minireview (p28-29)

Comment 2

A separate section highlighting the advantages of using blood-borne biomarkers over the CSF ones, as mentioned by authors in abstract, would help in an improved understanding of the main idea behind this review.

Our response

We have added a separate section highlighting the advantages of using blood-borne biomarkers over the CSF ones (page 16, lines 384-page 17, lines 408) as follows, “Advantages of blood biomarkers over CSF biomarkers

CSF biomarkers for AD have been studied for more than 20 years, and many powerful markers have been identified for diagnosis, prognosis, or even prediction of the future onset of AD. The combined use of these CSF markers may largely improve the accuracy and sensitivity of AD diagnosis at the early stage. However, because obtaining CSF is invasive and may induce prognostic symptoms, physical examination using CSF samples to screen for the risk of AD in large populations of asymptomatic people is not practical.

Blood test indexes, such as blood cholesterol, triglyceride, high-density lipoprotein cholesterol, and low-density lipoprotein cholesterol have been widely used for predicting the risks for arteriosclerosis and cerebrovascular and cardiovascular diseases in healthy and asymptomatic populations. However, a safe, less invasive, and readily accessible blood marker for AD diagnosis or predicting the risk of AD is still not at clinical use stage. In annual physical examinations, blood samples were routinely collected in large and healthy population from middle age to advanced age. The number and scale of blood samples will take great advantage over CSF samples of developing blood biomarkers for AD. In the past 15 years, many studies on blood markers for AD diagnosis, prognosis, and prediction have been performed, and some biomarkers have emerged as candidates for less invasive blood markers for AD (Table 1). For example, recent studies from different groups suggest that decreased blood Aβ42 and increased blood p-tau 181 may reflect brain amyloid deposition and neurofibrillary tangles, respectively, at the early stage of AD. The changes of some plasma proteins, lipids, Aβ43, and flotillin in the blood samples from AD patients are also needed to be confirmed by different groups. Nevertheless, using blood-borne biomarkers to make clear AD diagnosis or prognosis will be available in the near future. ”

Comment 3

Have there been any clinically relevant study with blood biomarkers in AD that can be added in the text to highlight possible use of blood biomarkers in a clear AD diagnosis and treatment since all others have failed till now?

Our response

To our knowledge, there is still no successful clinically relevant study with blood biomarkers for clear AD diagnosis and treatment.

Comment 4

A simple figure describing the disease pathology of AD in terms of amyloid precursor protein processing, Aβ aggregation, tau neurofibrillary tangles and secretory components involved in the diseases etiology would increase the overall value of the manuscript.

Our response

We have included a figure (Figure 1) describing the disease pathology of AD and associated biomarkers.

Reviewer 2 Report

Zou et al in this mini review have tried to provide a comprehensive review on the Current biomarkers for Alzheimer’s disease. The mini review is well written and has message.

The authors do provide the reasons for the requirement of this mini review and have current biomarkers available. 

The authors need to provide how the current caveats in research identifying the biomarkers could be overcome. But have not explained how this mini review would add value to the current set of information.

The authors also need to provide a future direction in the mini review that would enhance the current research in discovering targets and therapy.

Author Response

Comment 1

The authors need to provide how the current caveats in research identifying the biomarkers could be overcome. But have not explained how this mini review would add value to the current set of information.

Our response

We have added new description and recent identified blood biomarkers for AD in the abstract (page 2, lines 39-43) as follows, “Elements of the core pathogenesis of AD in blood, including Aβ42, tau proteins, plasma proteins or lipids have shown their usefulness and capabilities in AD diagnosis. We also highlight some novel identified blood biomarkers (including Aβ42/Aβ43, p-tau 181, Aβ42/APP669-711, structure of Aβ in blood, and flotillin) for AD.”

Comment 2

The authors also need to provide a future direction in the mini review that would enhance the current research in discovering targets and therapy.

Our response

We have added a separate section highlighting the advantages of using blood-borne biomarkers over the CSF ones (page 16, lines 384-page 17, lines 408) as follows, “Advantages of blood biomarkers over CSF biomarkers

CSF biomarkers for AD have been studied for more than 20 years, and many powerful markers have been identified for diagnosis, prognosis, or even prediction of the future onset of AD. The combined use of these CSF markers may largely improve the accuracy and sensitivity of AD diagnosis at the early stage. However, because obtaining CSF is invasive and may induce prognostic symptoms, physical examination using CSF samples to screen for the risk of AD in large populations of asymptomatic people is not practical.

Blood test indexes, such as blood cholesterol, triglyceride, high-density lipoprotein cholesterol, and low-density lipoprotein cholesterol have been widely used for predicting the risks for arteriosclerosis and cerebrovascular and cardiovascular diseases in healthy and asymptomatic populations. However, a safe, less invasive, and readily accessible blood marker for AD diagnosis or predicting the risk of AD is still not at clinical use stage. In annual physical examinations, blood samples were routinely collected in large and healthy population from middle age to advanced age. The number and scale of blood samples will take great advantage over CSF samples of developing blood biomarkers for AD. In the past 15 years, many studies on blood markers for AD diagnosis, prognosis, and prediction have been performed, and some biomarkers have emerged as candidates for less invasive blood markers for AD (Table 1). For example, recent studies from different groups suggest that decreased blood Aβ42 and increased blood p-tau 181 may reflect brain amyloid deposition and neurofibrillary tangles, respectively, at the early stage of AD. The changes of some plasma proteins, lipids, Aβ43, and flotillin in the blood samples from AD patients are also needed to be confirmed by different groups. Nevertheless, using blood-borne biomarkers to make clear AD diagnosis or prognosis will be available in the near future. ”